# Contrastive Decision Transformers

**Sachin Konan**[*]
Computer Science
Two Sigma Investments, LP
sachin@twosigma.com

**Esmaeil Seraj**
Interactive Computing
Georgia Institute of Technology
eseraj3@gatech.edu

**Matthew Gombolay**
Interactive Computing
Georgia Institute of Technology
matthew.gombolay@cc.gatech.edu

**Abstract:** Decision Transformers (DT) have drawn upon the success of Transformers by abstracting Reinforcement Learning as a target-return-conditioned, sequence modeling problem. In our work, we claim that the distribution of DT's target-returns represents a series of different tasks that agents must learn to handle. Work in multi-task learning has shown that separating the representations of input data belonging to different tasks can improve performance. We draw from this approach to construct ConDT (Contrastive Decision Transformer). ConDT leverages an enhanced contrastive loss to train a return-dependent transformation of the input embeddings, which we empirically show clusters these embeddings by their return. We find that ConDT significantly outperforms DT in Open-AI Gym domains by 10% and 39% in visually challenging Atari domains. Additionally, ConDT shows promising application to robot learning by outperforming DT by 20% in the Adroit Robotic HandGrip Experiments.

**Keywords:** Reinforcement Learning, Decision Transformers, Contrastive Loss

## 1 Introduction

Decision-making is a reasoning process resulting in the selection of a belief or a course of actions that leads to a desired outcome [1]. Decision-making has been conventionally approached by Reinforcement Learning (RL) methods [2] in robotics [3, 4, 5], game-playing [6, 7, 8], and multi-agent collaboration [9, 10, 11, 12, 13, 7, 14, 15], as well as by Learning from Demonstration (LfD) methods [16, 17]. In RL, decision-making is framed as learning a mapping from a state observation to an action that maximizes the cumulative discounted environment rewards (i.e., return), and has seen success with traditional methods that fit value functions [18] or compute policy gradients [19]. Recently, Chen et al. [20] introduced the Decision Transformer (DT) to the realm of RL to draw from the simplicity and scalability of transformer architectures [21]. DT abstracts the decision-making process in RL as a sequence modeling problem and attempts to learn a *return-conditioned* state-action mapping [20]. The return-conditionality means that given a history of return-state-action tokens, such that the last token represents the desired return at the current-state, the DT predicts the action required to achieve this desired return. DT achieved promising performance in many decision-making problems [20, 22], but there still exists a sizable margin for transformers to improve in the realm of RL as shown in the results of Chen et al. [20] where DT matched or performed worse than existing offline RL baselines in a bevy of tasks.

In this work, we hypothesize that integrating Contrastive Representation Learning (CRL) into the DT architecture will help DT learn a higher quality state-to-action mapping (i.e., policy) and achieve better convergence results. The objective of CRL is to cluster data representations with respect to their respective classes in order to strengthen the discriminability of the representations generated by Neural Networks (NN) [23, 24]. The power of CRL can be seen from its success in a large

---

[*]Research was conducted while as an undergraduate at Georgia Tech.

range of learning problems [23, 25]. Therefore, we investigate decision transformers trained with a contrastive loss, since DT is an inherently *return-conditioned* system (i.e., the predicted optimal action depends on the magnitude of user-defined returns at the current time-step).

The distribution of the target returns with which the DT is conditioned upon represents a distribution of sub-tasks with which the DT must learn to handle. To simultaneously handle the conflicting gradients of differing tasks, we posit that modifying the DT architecture to include a contrastive loss and an input transformation layer to transform the input data into mutually separated sub-spaces of the input-dimension. We posit that this will implicitly help with the performance of DTs. We accomplish this process by introducing Contrastive Decision Transformers (ConDT), which enables learning discriminable input-space embeddings for DT by categorizing the embeddings and the sub-task to which they belong. Our new modified architecture allows DT to more easily choose the optimal action corresponding to the current state, desired return, and history of information. Our results in Atari and Open-AI-Gym environments show consistent improvements over the conventional DT by achieving significantly higher return-performance. Our contributions are as follows:

1. We propose ConDT, a modified, contrastive DT architecture and training procedure that utilizes CRL and return-dependent embedding-space transformations to effectively discriminate state and action embeddings belonging to different return classes.

2. We propose an enhanced CRL objective, which we call the $\mathcal{L}_{\text{SimRCRL}}$ loss, for optimizing representational distance of state and action embeddings. We show that our $\mathcal{L}_{\text{SimRCRL}}$ results in better *separation* of these embeddings in representation space.

3. We demonstrate that ConDT improves return performance by an average of **10%** in Open-AI Gym, **39%** in Atari, and **20%** in the Adroit Robotic Handgrip experiments. These notable gains indicate the strength of applying CRL to the transformers' input embeddings.

## 2 Related Work

RL has seen great success in decision-making problems in robotics [3, 4], game-playing [6, 7], and multi-agent collaboration [15, 10, 11, 26]. For improved sample-efficiency, researchers have studied offline RL [4], in which the agent learns from a pre-recorded dataset rather than environment interactions. Offline RL methods are highly applicable to robot learning tasks with effective performance and has been leveraged in robotic hand and object manipulation tasks in prior work [27, 28]. Q-learning is one of the most successful offline methods [18, 29], and with other methods, has benefited from observing a history of states and actions, rather than just the last state [30]. We refer to this "history" as temporal context. Recurrent-based architectures [31] were generally used with RL to solve domains (i.e., games) with temporal context, and with the introduction of transformers [21, 32], researchers have investigated replacing RNNs with transformers in offline RL [33, 34, 20]. In the following, we discuss the latest applications of transformers to RL with respect to ConDT.

**Transformers in RL –** Chen et al. [20] introduced Decision Transformer (DT) as a replacement to recurrent-based RL architectures for cases when temporal context is known. Additionally, Janner et al. [35] introduced Trajectory Transformer as a long-term planning framework for RL. In this work, we propose a modular input-embedding space transformation and contrastive procedure (ConDT), which was tested on DT, but applicable to other transformers. ConDT achieves success because its methods effectively categorize the distribution of returns as distinguishable sub-tasks. This is notably distinctive from Multi Game DT (MGDT), which augment the input sequence of DT with observations from multiple-games showing that a single DT can generalize to multiple different games. MGDT trains a DT to solve multiple games, while ConDT trains a DT to better solve the sub-tasks within a single game.

**Contrastive Learning for Transformers –** Contrastive Learning has been applied to RL, contrasting representations in the state space [24, 36, 37] and the action space [38]. Liu et al. [39] recently introduced Return-Based Contrastive Learning (RCRL) which applies a contrastive loss across the return-space in RL. Contrastive methods for transformers generally take advantage of the InfoNCE loss [24], which has seen success in auto-regressive tasks. The InfoNCE loss has been successfully applied to multi-modal video Transformers [40] and bi-directional language Transformers [23]. These methods are relevant to non-RL domains, but our method, ConDT, attempts to apply RCRL to the input embedding layers of DT, because DT's architecture is inherently return-conditioned.

**Pre-Training Objectives for Transformers –** Transformers, in both vision and NLP applications, have benefited from pre-training [41, 42]. These works benefit from either introducing pre-training tasks related to the original objective [42], whereas other works perform pre-training on a different training set and then fine-tune on the main set [43, 44]. ConDT builds off the intuition of the former works, but utilizes a return-based contrastive objective to train the input-embedding space, unlike the pretraining techniques introduced by Devlin et al. [42].

## 3 Preliminaries

**Problem Formulation –** We consider the problem of learning in a Markov Decision Process (MDP) defined by the four-tuple: $\langle \mathcal{S}, \mathcal{A}, \mathcal{P}, \mathcal{R} \rangle$, where $\mathcal{S}$ represents the state-space and $\mathcal{A}$ represents the action space. A particular state and action at a time-step, $t$ is represented by $s_t, a_t$, respectively. $\mathcal{P}$ represents the transition function $0 \leq P(s_{t+1}|s_t, a_t) \leq 1$, and $\mathcal{R}$ represents the reward function and the reward is defined as: $r_t = R(s_t, a_t)$. In this paper, we follow convention of the original DT and define return, $(g_t)$, as the non-discounted rewards-to-go: $g_t = \sum_t^\infty r_t$. As stated earlier, DT is an offline RL method where the learner only from some fixed limited dataset (i.e., environment trajectories) instead of obtaining data via environment interactions. Offline RL is known to be more challenging as it removes an agent's ability to explore [20].

**Decision Transformer (DT) Objective –** DT [20] takes as input a sequence of three-tokens: $(\langle g_{t-K}, s_{t-K}, a_{t-K} \rangle, \cdots, \langle g_t, s_t, a_t \rangle)$, where $K$ is the game-specific context length. DT encodes each token into an embedding and adds a positional encoding to each embedding. The embeddings are then fed into the GPT-2 Causal Transformer [45] where an attention mechanism is applied to predict a left-shifted version of the input: $(\langle \hat{s}_{t-K}, \hat{a}_{t-K}, \hat{g}_{t-K+1} \rangle, \cdots, \langle \hat{s}_t, \hat{a}_t, \hat{g}_{t+1} \rangle)$. During inference, $a_t$ is unknown and therefore, $a_t$ is a null-token. For training, and for inference, the only relevant output token is $\hat{a}_t$, which represents the model's predicted action that will maintain the return in the current-state. Considering the true-optimal action as $a_t$ and predicted action as $\hat{a}_t$, the training objective of DT in discrete and continuous environments can be shown as in Eq. 1.

$$\mathcal{L}_{\text{DT}} = \begin{cases} \|\hat{a}_t - a_t\|^2, & \text{if continuous,} \\ -a_t \log(\hat{a}_t[a_t]), & \text{if discrete, } a_t \text{ is an integer and } \hat{a}_t \in \mathcal{A} \end{cases} \quad (1)$$

**Return-Based Contrastive Learning (RCRL) Objective –** RCRL performs contrastive representation learning on state-action embeddings [39]. In any offline dataset, we are given a series of return-state-action tuples, $(g_t, s_t, a_t)$. Returns will range from $[g_{min}, g_{max}]$, which we will split into $L$ distinct return buckets, $\forall_{k \in L} b_k : [b_k(\text{lower}), b_k(\text{upper})]$, where $b_k$ refers to the $k$-th return bucket that ranges from $b_k(\text{lower})$ to $b_k(\text{upper})$. RCRL adds a discriminator, $\mathcal{D}(x, y)$, to the network architecture, which returns a scalar indicating the degree of correlation between embeddings $x$ and $y$. The RCRL loss (Eq. 2) samples a batch of $\mathcal{B}$ anchor state-action embeddings, $z_{ah}$, whom each have some bin designation, $b_{ah}$, as well as $\mathcal{B}$ positive embeddings, $z_p$, whom have bin designation $b_p$ s.t. $b_p = b_{ah}$, and also $\mathcal{B}$ embeddings, $z_n$, whom each have bin designation $b_n$ s.t. $b_n \neq b_{ah}$.

$$\mathcal{L}_{\text{RCRL}} = \frac{1}{2\mathcal{B}} \sum_{z_{ah}, z_p, z_n} \left( \left( \mathcal{D}(z_{ah}, z_p) - 1 \right)^2 + \left( \mathcal{D}(z_{ah}, z_n) \right)^2 \right) \quad (2)$$

## 4 Methodology

**Overview –** The full architecture of the proposed Contrastive Decision Transformer, ConDT, is shown in Fig. 1. Contrary to the original DT [20], we create a sub-space transformation layer that represents a return-dependent transformation of the state and action embedding spaces. The transformation layer is trained by our modified contrastive objective, the SimRCRL loss introduced in Eq. 4, to discriminate state-action embeddings depending on target return.

Next, we present a theoretical motivation for our following sections on return-based sub-space transformations and our contrastive learning objective, SimRCRL.

**Motivation and Insight –** Consider a simplistic reduction of the problem formulation, wherein we are given a memory bank $\mathbb{B}$ containing a set of three-tuples $(g'_t, s'_t, a'_t)$ where the distribution of returns is approximately uniform. Now, given some $g_t$ and $s_t$, what is the optimal $a_t$ that could achieve this $g_t$? A simple search-based solution could use $k$-nearest neighbors, whose complexity

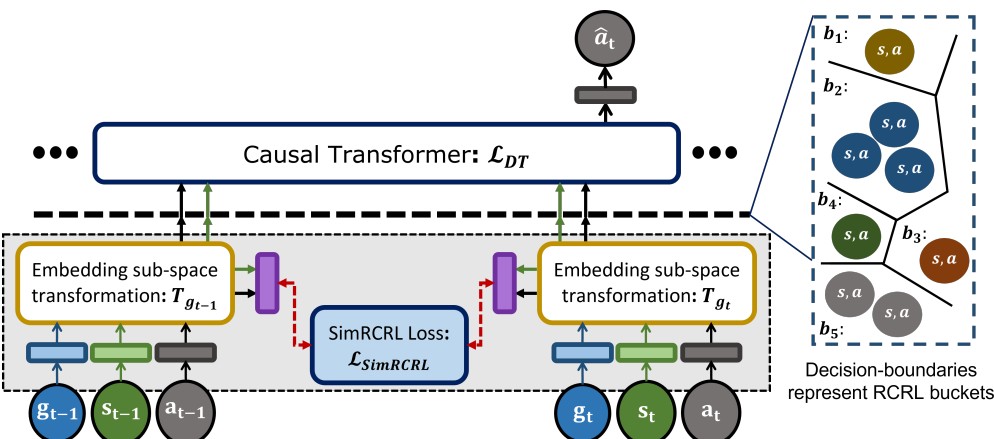

Figure 1: The architecture of the proposed Contrastive Decision Transformer. We create a sub-space transformation layer that represents a return-dependent transformation of the state and action embedding spaces. The transformation layer is trained by our modified contrastive objective, the SimRCRL loss in Eq. 4, to discriminate state-action embeddings depending on target return, as shown in the box on right. The purple box is a state and action embedding compression layer.

is $O(|\mathbb{B}|(|\mathcal{G}| + |\mathcal{S}|)$, where $|\mathcal{G}|$ is the size of the return-space and $|\mathcal{S}|$ is the size of state-space. Upon finding a subset of tuples that match the return, $g_t$, and state, $s_t$, their associated $a'_t$ can be chosen as the optimal action. However, imagine if the memory bank was organized by $g'_t$ before-hand, i.e $\{g'_t : [(s'_t, a'_t), \cdots], \cdots\}$. We can index the memory bank by $g_t$ in $O(1)$, resulting in $|\mathbb{B}|/|\mathcal{G}|$ state-action 2-tuples, and apply the $k$-NN across the state-space of the 2-tuples to find the optimal $a'_t$. Therefore, restructuring the memory bank reduces complexity to $O(|\mathbb{B}||\mathcal{S}|/|\mathcal{G}|)$. This is a simplistic reduction of the problem formulation, because transformers effectively search through many combinatorial permutations of $(g'_t, s'_t, a'_t)$, dependent on the context-length; however, this analysis motivates the benefit of pre-indexing the data by return. Since our data is high-dimensional, we turn to the transformations and contrastive methods in the following sections to effectively "index" the state and action embeddings by return.

**Return-Based Embedding Sub-Space Transformation –** As stated in Section 1, the distribution of potential, conditionable returns represents a distribution of sub-tasks which DT must solve. Cheung et al. [46] proposed several transformations that could be applied to the input data, such that data corresponding to different tasks existed in orthogonal sub-spaces of $\mathbb{R}^N$, where $N$ is the dimension of the input, $x$. They considered a task-dependent context transformation, $C_k$, where $k \in L$ and $L$ is the distribution of possible tasks. For a single-layer, linear network $(\mathcal{W})$, the output corresponding to the $k$-th task can be written as $y_k = \mathcal{W}(C_k x)$.

In our formulation, we denote the transformation $C_k$, as $T_{g_t}$ which indicates a return-dependent, $g_t$, transformation applied to each embedding space. We note that, here we remove $g_t$ from DT's input set of tokens, since the $T_{g_t}$ should effectively *encode* $g_t$ into the representations of $s_t$ and $a_t$, the $z^s_t$ and $z^a_t$, respectively. Therefore, we modify the input to the GPT [45] portion of DT as follows:

$$\text{GPT}\left([T_{g_{t-k}} z^s_{t-K}, T_{g_{t-k}} z^a_{t-K}, \cdots T_{g_t} z^s_t, T_{g_t} z^a_t]\right) \rightarrow [s_{t-K}, a_{t-K}, g_{t-K+1}, \cdots s_t, a_t, g_{t+1}] \quad (3)$$

Now, we need to choose an optimal transformation function, $T_{g_t}$. To this end, we follow the static rotation-based approach proposed by Cheung et al. [46]. Following the return-bin notation of the RCRL objective in Section 3, we can generate the relevant rotation matrix, $R_{b_k}$, associated with some given return bucket-label, $b_k$, by sampling some orthogonal matrix from the Haar Distribution [46] for each possible bucket. Thus each sequence of $(g_t, s_t, a_t)$ tokens can appropriately be transformed by defining $T_{g_t} = R_{b_k}$, s.t. $k$ is the return bucket index that $g_t$ belongs to. Additionally, we design $T_{g_t}$ to be a learnable transformation, by considering $T_{g_t}$ as a diagonal matrix whose entries are populated by the embedding generated by $g_t$. In this case, $T_{g_t}$ is simply equivalent to the vector-product of the embedding generated by $g_t$ and the embeddings generated by $s_t$ and $a_t$. However, with a learnable $T_{g_t}$, there is no guarantee that the transformed embeddings be distant in representation space. Accordingly, we develop a contrastive objective for $T_{g_t}$.

**Contrastive Learning of the Embedding Transformation –** We explore contrastive learning as a direct optimization objective to train $T_{g_t}$ to push state and action embeddings belonging to different $b_k$ to be more distant in representation space. The was also the motivation behind the RCRL loss [39], introduced in Section 3, which claims that training a discriminator on pairs of state-action embeddings will cause the embeddings to grow distant/together in representation space. This claim was supported through an analysis of the cosine-similarity change over time [39], although, in our experiments, we found that optimization of the discriminator required by RCRL **did not always** strictly achieve this result.

Here, we improve upon the RCRL objective, in Eq. 2, by first sampling a batch, $\mathcal{B}$, of bins from $L$. Then, for each bin we sample two sets of $(g_t, s_t, a_t)$ tuples, where the first is considered the the *anchor* and the second the *positive* sample associated with the anchor. Therefore the loss considers a total of $2\mathcal{B}$ samples. For a given anchor and its associated positive, the rest of the other anchors/positives serve as negative samples (i.e., $2N - 2$), with which we may contrast the embeddings upon. We define the strict-RCRL objective as the normalized-temperature-cross entropy loss of the state-action embeddings, whose construction is drawn from SimCLR [47]. We call our objective SimRCRL, which is defined as in Eq. 4, where the compressed representation of the $i$-th anchor, which is a state-action embedding, is indicated by $z_{ah}^i$ and its associated positive pair embedding as $z_p^i$, $\mathbb{1}(i,j)$ is the indicator function that is 0 when $i = j$ and 1 if not, and $\tau$ as the temperature hyper-parameter. $\mathcal{B}_C$ is the batch-size used in $\mathcal{L}_{\text{SimRCRL}}$.

$$\mathcal{L}_{\text{SimRCRL}} = \sum_{i=0}^{\mathcal{B}_C} -\log\left(\frac{\exp((z_{ah}^i \cdot z_p^i)/\tau)}{\sum_{j=0}^{\mathcal{B}_C} \mathbb{1}(i,j)[\exp((z_{ah}^i \cdot z_{ah}^j)/\tau) + \exp((z_{ah}^i \cdot z_p^j)/\tau)]}\right) \quad (4)$$

Now, putting everything together, our total contrastive loss for training ConDT, $\mathcal{L}_{\text{ConDT}}$, is the sum of DT's objective and our improved contrastive objective, weighted by a $\beta$ parameter that weighs how much the contrastive objective should affect the DT during training. $\mathcal{L}_{\text{ConDT}}$ is shown in Eq. 5. During experimentation, we consider pre-training the embeddings layers with $\mathcal{L}_{\text{SimRCRL}}$, followed by training the entire DT with a slowed learning rate on the embedding layers and $\mathcal{L}_{\text{DT}}$ loss.

$$\mathcal{L}_{\text{ConDT}} = \mathcal{L}_{\text{DT}} + \beta * \mathcal{L}_{\text{SimRCRL}} \quad (5)$$

## 5 Empirical Evaluation

**Baselines** - We tested our improvements upon the original DT architecture Chen et al. [20] by testing the below five methods. For each experiment, we examine the following methods where (1) is the original baseline DT [20] architecture, (2-3) are ablations that study the impact of learnable vs fixed return-transformations of the state and action embeddings, and (4-5) study the combination of (1) and (3) with the $\mathcal{L}_{\text{SimRCRL}}$ contrastive objective:

1. DT – Baseline DT proposed by Chen et al. [20]
2. DT+Rot – Baseline DT with *fixed*-rotation embedding sub-space-transformation
3. DT+Prod – Baseline DT with *learnable* input-embedding transformation
4. ConDT w/o Prod (Ours) – Baseline DT trained with our $\mathcal{L}_{\text{ConDT}}$ loss
5. ConDT (Ours) – DT+Prod trained with our $\mathcal{L}_{\text{ConDT}}$ loss

**Evaluation Environments –** We empirically validate the enhanced performance of ConDT against the baselines in several decision-making domains, including Atari 2600, Open-AI Gym, and the Adroit Handgrip Environments. For environment descriptions and details, please refer to the provided supplementary material. Next, we present our evaluation results and investigative studies of the learned representations. We note that we publicly provide our code-base (including ConDT implementation and the baselines) at https://github.com/CORE-Robotics-Lab/ConDT.

### 5.1 Results, Ablation Studies, and Discussion

**Open-AI Gym –** We first measure return performance across three standard Open-AI Gym locomotion domains: hopper, halfcheetah, and walker2d. We did not test in the Reacher domain, as done in DT [20], since the dataset for this domain was not publicly available. Agents are tasked with

achieving human-level return performance (i.e., 3600 in hopper, 12000 in halfcheetah, and 5000 in walker2d for results in Table 1). In Gym and Atari, we report the mean, $\mu$, and standard error, $SE$, of the return achieved in three random seeds.

We train the agents using the *Medium* (M.), *Medium-Replay* (M.R.), and *Medium-Expert* (M.E.) datasets provided by Fu et al. [48]. These datasets differ in size and the policies used to generate them. Thus, we tested on all three variations to study ConDT's robustness. Details of the distinction between these datasets can be found in the supplementary and in the original DT paper [20].

| Dataset | Env. | DT [20] | DT+Rot | DT+Prod | **ConDT w/o Prod** (Ours) | **ConDT** (Ours) |
|---------|------|---------|--------|---------|---------------------------|------------------|
| M. | hopper | 2097±38.9 | 1397±25.8 | 2330±33.2 | 2293±39.3 | **2403**±30.6 |
| | halfcheetah | 4981±28.1 | 5035±36.8 | 5038±14.9 | 5020±11.7 | **5060**±**11.7** |
| | walker2d | 3466±56.3 | 3453±55.9 | 3521±30.5 | **3609**±**44.8** | 3545±49.4 |
| M.R. | hopper | 2393±43.4 | 13.6±0.07 | 2959±16.3 | 2869±29.2 | **3076**±**27.7** |
| | halfcheetah | 4070±52.4 | 703.4±39.4 | 4791±30.6 | 4491±35.8 | **4803**±**36.2** |
| | walker2d | 2988±114 | 968.3±96.1 | **3717**±**65.7** | 3322±77.4 | 3332±76.6 |
| M.E. | hopper | 3133±61.0 | 1528±65.5 | 3354±56.9 | 3505±26.3 | **3571**±**8.49** |
| | halfcheetah | 10779±29.2 | 8956±189 | 11021±62.1 | 10821±49.5 | **11218**±**14.5** |
| | walker2d | 4941±12.9 | 1195±73.4 | 4951±2.61 | 4850±35.6 | **5011**±**1.44** |

Table 1: Baseline Gym test results for medium (M), medium-replay (M.R.), and medium-expert (M.E.) datasets (as described in [20]) of hopper, half-cheetah, and walker2d, shown as $\mu \pm$ SE

As shown in Table 1, DT+Prod and ConDT w/o Prod exceed the performance of DT, while ConDT exceeds or matches DT+Prod and ConDT w/o Prod in 8/9 tasks. This shows that learnable, return-based transformations benefit return performance, which can be further enhanced with the contrastive training objective. DT+Rot either matches or performs slightly worse than DT, although this trend dramatically falters across the datasets in *Medium-Replay*. We hypothesize the reason for the significant performance dip is because the return distribution in medium-replay is much more left-skewed than medium or medium-expert. This would cause the inputs to DT+Rot to only be transformed with $R_{b_k}$ where $b_k$ corresponded to bins in the lower range of the return-distribution, and thus harm inference when larger returns were provided to DT+Rot.

**Atari 2600 –** We also tested on Atari 2600 environments [49], which contains of suite of classic Atari Games that are generally considered challenging due to the high dimensionality of the observation space ($(210 \times 160)$ RGB image) and time-delayed credit assignment of rewards. We train each baseline on 1% of all samples in the DQN-replay dataset provided by Agarwal et al. [29], representing 500K out of the 50 million transitions observed by an online DQN. We tested all the baselines across four domains, Breakout, Qbert, Seaquest, and Pong, and results are in Table 2.

| Env. | Target $g_t$ | DT [20] | DT+Rot | DT+Prod | **ConDT w/o Prod** (Ours) | **ConDT** (Ours) |
|------|--------------|---------|--------|---------|---------------------------|------------------|
| Breakout | 90 | 48.8±6.69 | 42.7±2.08 | 70.4±4.73 | 67.8±4.73 | **71.1**±**2.46** |
| Qbert | 14000 | 3763±348 | 6877±302 | 3677±303 | **10735**±**346** | 6432±314 |
| Pong | 20 | 16.8±0.57 | 11±0.59 | 16.7±0.29 | 13.2±0.59 | **17.9**±**0.48** |
| Seaquest | 1150 | 948±34.7 | 880±32.1 | 1018±32.7 | **1364**±**29.0** | 1250±27.3 |

Table 2: Baseline Atari test results for Breakout, Qbert, Pong and Seaquest, shown as $\mu \pm$ SE.

Causal attention allows GPT [45] to understand the relation between each token and its preceding sub-sequence. Therefore, the removal of the return tokens from the input to GPT, as is the case in DT+Rot, DT+Prod, and ConDT, gives rise to the concern of whether the removal will have a negative effect on the performance of the overall DT. As shown in Table 2, even without learnable return-transformation, DT+Rot performs only slightly worse than DT across 3/4 of the Atari domains and doubles the performance of DT in Qbert, indicating return has successfully been encoded into the state and action embeddings. With learned transformations, DT+Prod and ConDT w/o Prod outperform DT in 3/4 of the experiments, except in Pong, where DT+Prod matches DT's performance, but most notably, in Qbert, ConDT w/o Prod nearly triples the performance of DT. We hypothesize that ConDT w/o Prod outperforms ConDT in 2/4 tasks, because its causal attention allows it to relate return embeddings from any preceding time-step to a specific state, action embedding, whereas in ConDT, return embeddings are only related to the state, action embeddings at the same specific timestep.

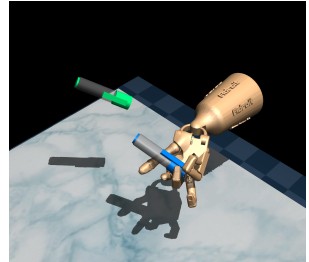 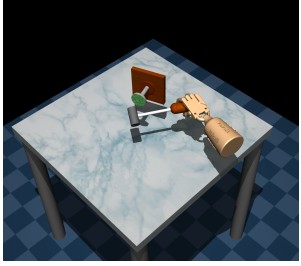 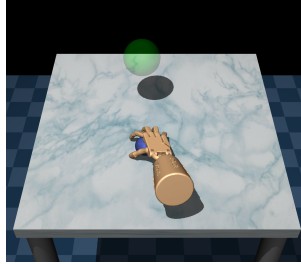

(a) Pen: orientate a pen     (b) Hammer: hammer a nail     (c) Relocate: move blue ball

Figure 2: The Adroit Robotic HandGrip, a 24 degree-of-freedom simulated robot hand domain, including three difficult manipulation tasks (i.e, Pen, Hammer, and Relocate).

**Robotic HandGrip Manipulation –** Previous works have studied the application of offline RL to robotics [27, 28]. In our work, to study the applicability of ConDT to robot learning, we investigate the Adroit Robotic HandGrip Environments [27], wherein a 24 degree-of-freedom robotic hand is trained to hammer a nail (i.e., Hammer), orient a pen (i.e., Pen), and relocate an object (i.e., Relocate). Fig. 2 shows a sample demonstration of these complex manipulation tasks. We detail ConDT's performance gains in Table 3, and provide visualizations of the trained policies in the supplementary material. We also provide a video demonstration of the simulated Adroit Robotic HandGrip executing the learned policies by DT and ConDT for each task as supplementary material. The demo showcases the superior policy learned through our ConDT over the base DT.

| Env. | Target $g_t$ | DT [20] | DT+Prod | **ConDT w/o Prod** (Ours) | **ConDT** (Ours) |
|---|---|---|---|---|---|
| Pen | 4000 | 1465±159 | 1883±192 | 1560±162 | **2016±179** |
| Hammer | 12000 | 14036±446 | 16036±187 | 16078±98 | **16364±210** |
| Relocate | 4500 | 4331±465 | 4362±477 | 4483±402 | **4526±210** |

Table 3: The Adroit Robotic HandGrip test results for ConDT and baselines, shown as $\mu \pm$ SE.

## 5.2 Investigating the Learned Representations

We hypothesize that our return-based transformation, $T_{g_t}$, of the input embedding-space with the $\mathcal{L}_{\text{ConDT}}$ objective helps distance the state and action embeddings based on the return-class they belong to. To support this hypothesis, we study how the distance in the state and action embeddings change over time. Additionally, we visually investigate clustering behavior in the embedding-space.

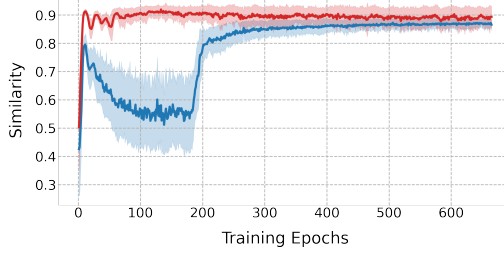 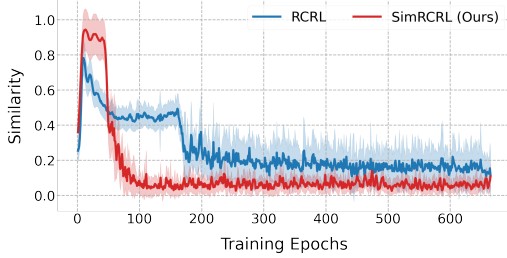

(a) Positive Similarity during Pre-Training     (b) Negative Similarity during Pre-Training

Figure 3: Comparing the loss convergence between $\mathcal{L}_{\text{RCRL}}$ and $\mathcal{L}_{\text{SimRCRL}}$

**Cosine Similarity Analysis –** For this analysis, we seek to understand how the representation distance of state-action embeddings from ConDT change over time w.r.t. two loss functions: $\mathcal{L}_{\text{RCRL}}$ and $\mathcal{L}_{\text{SimRCRL}}$. For this purpose, we utilize the cosine similarity as the distance metric [47]. Here a value of 1 represents parallel embeddings, and 0 represents orthogonal embeddings. Therefore, state-action embeddings of the same bins should desirably have similarity value close to 1 (i.e., high positive similarity or low *intra*-class distance) while representations belonging to different bins should have a similarity value close to 0 (i.e., low negative similarity or high *inter*-class distance).

During pre-training of ConDT with our modified $\mathcal{L}_{\text{SimRCRL}}$ objective, we sampled a fixed number of data samples across different return bins. Then we measure the average cosine similarity to samples within the same return bin and different return bins, which we call positive and negative similarity respectively in accordance with [47]. In Fig. 3a and Fig. 3b, we show the positive and negative similarities from the contrastive training of ConDT for the Atari-Breakout experiment, comparing $\mathcal{L}_{\text{SimRCRL}}$ with $\mathcal{L}_{\text{RCRL}}$. $\mathcal{L}_{\text{SimRCRL}}$ results in more stable convergence of the positive-similarity and negative-cosine similarity, whereas $\mathcal{L}_{\text{RCRL}}$ loss does not result in a monotonic increase of the positive-similarity, nor strict convergence to either 1 or 0 for positive and negative similarity. Therefore, our enhanced $\mathcal{L}_{\text{SimRCRL}}$ objective is more effective in achieving the transformation-based objective we formulated for application to DTs.

**Embedding-Space Visualization –** Here, we intend to visualize the state-action embedding-space for a comparison between DT and ConDT. The goal is to observe a clustering behavior as result of our embedding-space transformation with contrastive learning, such that the between-class distances would increase and the within-class distances would decrease between the embeddings. To visualize the joint state-action embedding space, we sampled a batch of ten return bins, with 30 samples per bin. Then, we generated the state-action embeddings of this input data for DT and ConDT, and created a 2D TSNE visualization of the embeddings, shown in Fig. 4. As shown, DT demonstrates no clustering behavior of the state-action embeddings based on return; points of the same color are scattered amongst points of different colors (i.e., colors indicate the return bins). However, for ConDT, points of the same color are clearly clustered together and have distance to points of different color, indicating that the contrastive embedding effectively pushes state-action embeddings of different returns apart, and similar returns together.

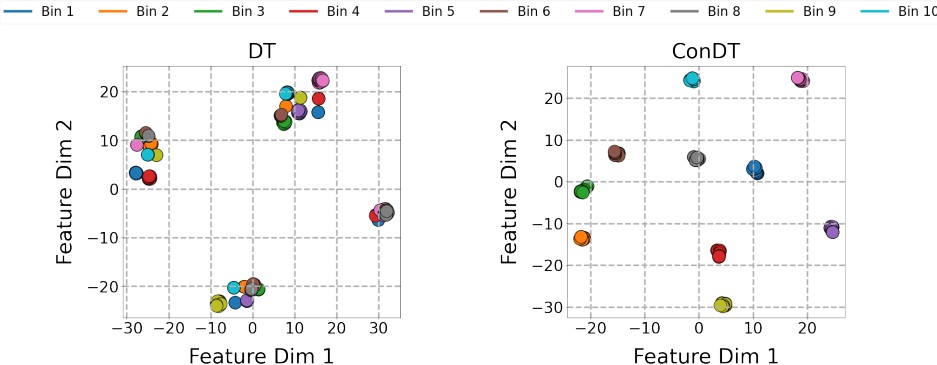

Figure 4: TSNE 2D visualization of state-action embedding-spaces for DT and ConDT.

## 5.3 Limitations

A limitation DT [20], which is shared in ConDT, is the size and distribution of returns in the offline data. As mentioned in the Open-AI Gym analysis, the left-skewed nature in the return distribution of medium-replay dataset harmed the performance of DT+Rot. The original DT [20] addresses this concern by sampling trajectories according to the magnitude of their cumulative return; thus, causing higher return trajectories to be sampled more often. Therefore, sampling techniques and the return distribution of the dataset can become a bottleneck in decision transformer methods.

## 6 Conclusion

Decision Transformer (DT) was one of the first introductions of transformers to RL, and while experimentation showed promise, there still lies a performance gap between DT and existing offline RL methods [20]. In this work, we hypothesized that performing a return-dependent transformation of the input embeddings to the DT can help enhance return performance. We proposed Contrastive Decision Transformer (ConDT). ConDT performs an embedding-space transformation of the state and action embeddings, where the transformation is trained using our enhanced $\mathcal{L}_{\text{SimRCRL}}$ objective, which we empirically show can better maximize distance in state-action embedding space. We verify the applicability to robot learning and strength of ConDT's return-based transformation and contrastive learning objective by testing across several Atari and Open-AI Gym domains, as well as multiple 24-DoF robotics handgrip tasks, in which we show significant performance gains.

**Acknowledgments**

This work is supported by the Naval Research Lab under grant N00173-21-1-G009. We also thank the Two Sigma for sponsoring travel to/from the conference for the first author. However we note that: "the views expressed herein are solely the views of the author(s) and are not necessarily the views of Two Sigma Investments, LP or any of its affiliates. They are not intended to provide, and should not be relied upon for, investment advice."

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
