# OpenReview forum: "Contrastive Decision Transformers"
_robot-learning.org/CoRL/2022/Conference — CoRL 2022 Poster_

### Official Review · Reviewer_bC6T · 2022-07-18

**Originality:** Good
**Technical Quality:** Good
**Clarity Of Presentation:** Good
**Impact:** 3

**Recommendation:**

Weak Accept: I recommend accepting the paper, but will not argue for my recommendation if the majority of other reviewers have a different opinion.

**Summary:**

This paper proposes a Contrastive Decision Transformer (ConDT) that improves DT by treating the bin of target returns as each task and separating the representation of state and action embeddings into clusters labeled by each return bin. Specifically, the paper introduces a transformation layer for each bin and introduces a contrastive loss that makes embeddings from each transformation be apart from each other. The proposed method is evaluated on OpenAI Gym locomotion tasks and Atari games.

**Issues:**

- Not enough, or entirely no relevance to robot learning. Motivation should be more clear in the context of robotics, and the experiments should more focus on robotics tasks. Analysis and ablations on Atari domain is very awkward to be seen in a CoRL submission.
- Presentation can be improved

**Quality Of The Limitations Section:**

Additional details required

**Reviewer Expertise:**

4: The reviewer is confident but not absolutely certain that the evaluation is correct

**Robotics Focus:**

Relevant but unlikely to deploy to hardware in near future

**Strengths And Weaknesses:**

Strengths
- Idea is intuitive and interesting. Especially the intuitive explanation on how the proposed idea of separating the representations can effectively reduce the search space that would enable DT to perform better is interesting.
- Experimental results are interesting with supporting ablations and analysis. Trends in ablations are not perfect, but it can be understood.

Weaknesses
- The main weakness I see from the paper is that the paper is somehow submitted to not-relevant conference. In CoRL guideline, it is stated that ```All CoRL submissions must demonstrate the relevance to Robot Learning through Intent—explicitly address a learning question for physical robots, or Outcome—test the proposed learning solution on physical robots.``` This concern is intensified when I see the experimental results on Atari games -- how this could be related to robot learning? Last year's CoRL guideline also states that experiments on OpenAI gym locomotion tasks (e.g., Cheetah) should be supported with **intent** for transferring to real robots but I couldn't see any argument on it. How does the method solve challenges that would arise when learning robots in real-world? Moreover, as far as I know, there's still no successful application of DT on real robots.
- Presentation can be improved -- Please do not assume that readers would know the details from the introduction. For instance, in Contribution point 2, readers would now know what *discriminator approach* and definitely would not know $\mathcal{L}_{\text{SimRCRL}}$ loss. It would be nice to provide high-level idea of your method in contributions without assuming the knowledge of your method that can be known after reading the method section.
- Eq 3: This is a question, how could the length of input and target sequence could be different in causal transformer architecture of GPT?

Minor comments
- line 26: \citet -> \cite
- line 30: DT's -> DT
- line 34: Reference is wrong. Why Multi-Game DT [12] is referred in the success of contrastive learning? Citing representative works like CPC, SimCLR, MoCo would be desirable.

**Summary Of Recommendation:**

I would like to suggest to reject that paper, because it seems that the paper is not relevant to the robot learning conference. The paper did not mention or show the intend to improve robot learning, which makes it difficult to see the paper is of interest to the CoRL community. Of course the paper could be augmented with additional robotics experiments, I think it needs a significant revision with excluding Atari experimental results and including new main results, ablations, and analysis, which makes it difficult to recommend the acceptance of the paper.

However, experimental results are still interesting, so I would like to recommend the authors to improve the presentation of the draft and then submit the paper to other machine learning conference whose topic includes generic offline RL method.

---

> ### Author Response · Authors · 2022-08-27
> **Author Responses to Reviewer bC6T**
>
> Thank you! We appreciate your time and dedication. Please find below our point-by-point responses to your comments:
>
> - **Applicability to Robot Learning**: We agree that including real robot experiments is a necessity and doing so is our intention. However, respectfully, we do not agree that our method has “*not enough, or entirely no relevance to robot learning*”, as mentioned by the reviewer. Our approach, ConDT, and other similar offline RL methods are relevant to real physical robot learning, just as LfD writ large are relevant (see [1-2] for instance). A key difference is that many LfD methods are in fact online and would thus likely need more environment rollouts. It is true that offline RL methods are restricted to the dataset and have limited exploration capability. Nevertheless, many robotics applications may not allow for exploratory actions, such as in robotic surgery. Therefore, we believe while offline RL and LfD are both highly applicable to robot learning, they can have different use-cases. As such, we added a new experiment on the Adroit Robotic HandGrip Environments, wherein a 24 degree-of-freedom robotic hand is trained on four different tasks, such as opening a door and hammering a nail. All tasks have a reward function proportional to the percentage of the task completed, so if the hand closes a door 50% of the way, half the maximum reward is given. Additionally, all trajectory was generated by a human operating an Adroit hand in each of the four scenarios. Therefore, success of ConDT in the simulation is in close correlation with its applicability to a real-world Adroit hand. Our new results (Quantitative Results Table 3 and Video Demonstration uploaded as a Supplementary File) demonstrate the effectiveness of our method and applicability of ConDT to real robots. We also added a new paragraph to the text describing how ConDT and similar offline RL methods are relevant methods for robot learning and how to deploy such methods on real physical robots.
>
> [1] Sinha, Samarth, Ajay Mandlekar, and Animesh Garg. "S4RL: Surprisingly simple self-supervision for offline reinforcement learning in robotics." Conference on Robot Learning. PMLR, 2022.
>
> [2] Chen, Letian, Rohan Paleja, and Matthew Gombolay. "Learning from suboptimal demonstration via self-supervised reward regression." Conference on Robot Learning. PMLR, 2021.
>
> - **Presentation Can be Improved**: Thank you! We revised the text to address the specific cases mentioned by the reviewer and elsewhere to improve the readability.
>
> - **How Could the Length of Input and Target Sequence be Different in Causal Transformer Architecture of GPT?** The length of the input and target sequences are always 1-1 in terms of length. One situation though where they are symbolically different is near the beginning and end of a trajectory. For example, say we are one step into the beginning of a trajectory, with some context length, K. When inputted into GPT, the input sequence is padded with K-1 input tokens, and 1 (G-S-A) token. At the output, we still predict K-tokens, but K-2 are zero tokens and two are the action predictions of the current timestep and the following timestep
>
> - **Minor Comments**: Thank you! We fixed all the typos and added correct citations
>
> ## Issues:
>
> - **Relevance**: Please see our response to your first comment above. Thank you
>
> - **Presentation**: Thank you! We revised the text to address the specific cases mentioned by the reviewer and elsewhere to improve the readability.
>
> We hope that our responses and added simulated robotic handgrip experiment, discussions, and details were satisfactory. If so, we would appreciate if the respected reviewer considered increasing their score. Also, if there’s any further questions or concerns, we would be more than happy to address them

---

> > ### Comment · Reviewer_bC6T · 2022-08-28
> > **Thank you for your response**
> >
> > I'm glad that authors added a more discussion on robot learning and more results on robotics tasks. My major concern on relevance to robot learning is almost addressed, so I'd like to increase my score to 'weak accept', conditioned on that authors will further revise the final draft to incorporate some notes below when accepted:
> >
> > - It is recommended to do more revision to the draft to put major focus on robot learning as a CoRL submission (or paper) in the Introducion, such as incorporating your discussion on the relation between offline RL and robotics in the first or second paragraph.
> > - Update the analysis and ablations on the robotics tasks instead of Atari results, because main analysis on Atari results seems a bit weird to me.

---

> > > ### Author Response · Authors · 2022-08-28
> > > **Thank you!**
> > >
> > > Yes, of course! We are grateful for the feedback and will make these changes when accepted. Thank you!

---

### Official Review · Reviewer_hRuX · 2022-07-31

**Originality:** Good
**Technical Quality:** Good
**Clarity Of Presentation:** Very Good
**Impact:** 4

**Recommendation:**

Weak Accept: I recommend accepting the paper, but will not argue for my recommendation if the majority of other reviewers have a different opinion.

**Summary:**

This paper proposes ConDT, which applies contrastive learning to decision transformer. Specifically, the author utilize return-based contrastive learning to optimize input embeddings such that state-action pairs with similar returns are clustered in the embedding spaces. The authors showed that ConDT improves the performance of DT by a large margin on standard offline RL benchmark.

**Issues:**

* Comparing training time.

* Comparison with other state-of-the-art offline RL algorithms

* Evaluation on more hard domains such as ant-maze

**Quality Of The Limitations Section:**

Limitations are addressed clearly

**Reviewer Expertise:**

4: The reviewer is confident but not absolutely certain that the evaluation is correct

**Robotics Focus:**

Relevant but unlikely to deploy to hardware in near future

**Strengths And Weaknesses:**

* Strengths

1. Simple yet effective method: given fixed dataset, contrastive learning can provide additional signals for NNs and the authors showed that it is useful for offline RL using decision transformer. The idea is simple and main components (decision transformer & return-based contrastive learning) are from existing works. However, its novel combination makes a big difference.

2. Large gains on standard benchmark: the authors showed that ConDT outperforms DT on standard offline RL benchmarks. However, it would be also nice if the authors could compare to other state-of-the-art offline RL algorithms

* Weaknesses

1. Computational overhead: it would be nice if the authors also can point out some drawback from the proposed method. For example, additional auxiliary loss can increase the training time.

**Summary Of Recommendation:**

I'd like to recommend "weak accept" because (1) the proposed idea is well-motivated and (2) ConDT is very effective even though core idea is simple.

---

> ### Author Response · Authors · 2022-08-27
> **Author Responses to Reviewer hRuX**
>
> Thank you! We appreciate your time and dedication. Please find below our point-by-point responses to your comments:
>
> - **Computational Overhead**: Thank you! As suggested, we added a new supplementary section and provided details regarding the training and evaluation times for ConDT and baselines. The new Section and discussions are added to the supplementary material, page 4, Section 4, Execution Gain, and Table 7. From our experiments, ConDT and ConDT with pretraining require 23% and 38% longer execution times across the different experiments. We believe this gain can be drastically reduced by parallelizing evaluation between epochs, and using a faster dataloader for the SimRCRL loss (which is the primary bottleneck with using our contrastive loss)
>
> ## Issues:
>
> - **Comparing Training Time**: Please see our response above. Thank you.
>
> - **Comparison with Other SOTA Offline RL Algorithms**: We agree that this addition could add to the value of the paper. We note that the original DT was compared to the SOTA offline RL algorithms, demonstrating better or comparable performance in all scenarios. Based upon this result, we choose the DT as the state-of-the-art. As such, we only focus on showing that our work can outperform the DT as the SOTA, which was already shown to be better than other offline RL baselines. We will include other baselines in future extensions of our work.
>
> - **Evaluation on Harder Domains**: We added a new experiment on a simulated robotic handgrip on different tasks. These tasks are harder and inherently more difficult to learn than the initial domains in the paper as they pose complex robot learning tasks. Our new results demonstrate the effectiveness of our method and applicability of ConDT to real robots
>
> We hope that our responses and added simulated robotic handgrip experiment, discussions, and details were satisfactory. If so, we would appreciate if the respected reviewer considered increasing their score. Also, if there’s any further questions or concerns, we would be more than happy to address them

---

### Official Review · Reviewer_werf · 2022-08-01

**Originality:** Fair
**Technical Quality:** Good
**Clarity Of Presentation:** Very Good
**Impact:** 3

**Recommendation:**

Weak Accept: I recommend accepting the paper, but will not argue for my recommendation if the majority of other reviewers have a different opinion.

**Summary:**

The paper presents a new decision transformer method that combines decision transformers with contrastive learning for better representation learning, The authors perform contrastive learning on the embedding after the return-dependent transformation based on prior works RCRL and show that such an addition to decision transformer can lead to performance boost on tasks in D4RL and Atari games.

**Issues:**

See the Strengths And Weaknesses section.

**Quality Of The Limitations Section:**

Limitations are addressed clearly

**Reviewer Expertise:**

4: The reviewer is confident but not absolutely certain that the evaluation is correct

**Robotics Focus:**

Relevant but unlikely to deploy to hardware in near future

**Strengths And Weaknesses:**

**Strengths:**

1. The idea of the method intuitively makes sense and is easy to understand. It is neat to see that decision transformers can leverage contrastive learning for better performance.

2. The results of the proposed method seem to improve over vanilla decision transformer on standard benchmarks.

**Weaknesses**:

1. The paper appears to be a bit incremental in terms of novelty. The method is heavily built upon prior works decision transformer and RCRL. It seems to me that the only difference is that the authors use SimCLR rather than the simple regression loss in RCRL as the contrastive loss.

2. The improvement over the decision transformer is rather marginal. In particular, the simple DT+Prod method can perform quite comparably to the proposed method, questioning the effectiveness of the contrastive loss.

3. Real-world experiments are regrettably missing.

**Summary Of Recommendation:**

Based on my comments in the Strengths And Weaknesses, I think the novelty and the results of the paper are not convincing enough. I would vote for a weak reject.

--------------------
post-rebuttal updates: thank you for the response! I appreciate that the authors added real-world experiments and offered clarifications. Therefore, I've updated my score to a weak accept.

---

> ### Author Response · Authors · 2022-08-27
> **Author Responses to Reviewer werf**
>
> Thank you! We appreciate your time and dedication. Please find below our point-by-point responses to your comments:
>
> - **Incremental Approach**: In terms of methodology, we agree that our methodology is an incremental improvement from DT and RCRL individually (most research is incremental); the fact that DT is return-conditioned and applying a return-based contrastive loss was the main part of the novelty which has not been addressed before. SimRCRL is also an improvement of RCRL because it directly optimizes our optimization objective of separating the distance between state-action embeddings. Despite the seemingly straightforward derivation of ConDT, our approach substantially outperforms prior work.
>
> - **Marginal Improvement Over the Decision Transformer**: We believe that there is a misunderstanding. The baseline mentioned by the respected reviewer (i.e., DT+Prod) is still created by us, as an incremental step towards the final variant of our method (i.e., an ablation) and is not the naïve DTi. ConDT variants (with the contrastive loss), which are built with and without the Prod step, substantially outperform the base DT. Moreover, ConDT outperforms DT in Gym by 10%; the reason that is not as high as the performance improvement in Atari is that DT was already performing close to human-target performance prior to any of our optimizations. Using ConDT just improved the performance of Gym experiments even more to bring return results closer to human-target returns.
>
> - **Missing Real-World Experiments**: Thank you! We agree that including real robot experiments is a necessity and doing so is our intention. As such, we added a new experiment on the Adroit Robotic HandGrip Environments, wherein a 24 degree-of-freedom robotic hand is trained on four different tasks, such as opening a door and hammering a nail. All tasks have a reward function proportional to the percentage of the task completed, so if the hand closes a door 50% of the way, half the maximum reward is given. Additionally, all trajectory was generated by a human operating an Adroit hand in each of the four scenarios. Therefore, success of ConDT in the simulation is in close correlation with its applicability to a real-world Adroit hand. Our new results (Quantitative Results Table 3 and Video Demonstration uploaded as a Supplementary File) demonstrate the effectiveness of our method and applicability of ConDT to real robots. We also added a new paragraph to the text describing how ConDT and similar offline RL methods are relevant methods for robot learning and how to deploy such methods on real physical robots
>
> We hope that our responses and added simulated robotic handgrip experiment, discussions, and details were satisfactory. If so, we would appreciate if the respected reviewer considered increasing their score. Also, if there’s any further questions or concerns, we would be more than happy to address them

---

### Official Review · Reviewer_JfJf · 2022-08-05

**Originality:** Fair
**Technical Quality:** Very Good
**Clarity Of Presentation:** Very Good
**Impact:** 3

**Recommendation:**

Weak Accept: I recommend accepting the paper, but will not argue for my recommendation if the majority of other reviewers have a different opinion.

**Summary:**

This work introduces a contrastive loss for decision transformers, this is motivated by the need for better state, action representations for different returns. A new loss is introduced to separate such embeddings and is compared to a reasonable baseline. Policy performance using such a embedding+loss is shown to be improved on several OpenAI and Atari benchmarks. Ablations show which components of this algorithm are useful. They include studies of the embeddings and their separation during training.

**Issues:**

* Repeat some of the comparison for different # of params of transformer models - fixed for all models within the comparison.
* Some comments in the limitations about a (s, a) embedding without historical context when the transformer itself has that context.
* Comment of limitations in different reward settings, a lot of settings use binary rewards would you still recommend this method.

**Quality Of The Limitations Section:**

Additional details required

**Reviewer Expertise:**

4: The reviewer is confident but not absolutely certain that the evaluation is correct

**Robotics Focus:**

Highly relevant to robotics but no hardware experiments

**Strengths And Weaknesses:**

Strengths:
* The paper is very clearly written: the algorithm is clear, repeatable and compared to strong baselines on convincing benchmarks.
* Additional ablations and studies of the embeddings confirm the intuitive motivation for such an alteration to DTs
* The improvements are significant.

Weakness:
* It would be a much more convincing results with some real robot experiments, it seems like it works in simulated setups though.
* One reason for using sequence models is to model non-markovian systems (quite common in robotics): returns cannot be determined by the current (s, a) alone and require historical (s, a). However, with the state-action embedding clustering this property is lost. It would interesting to see if there are cases of such setups where this markov-assumption clustering actually hurts.
* Some details of the transformer architecture would be useful: in the comparisons of DT and ConDT are they controlled for the same number of parameters? What would happen if the transformers for both models were much bigger, surely a big enough transformer can model this separation if it useful to the task.
* I assume there is one embedding for the state and another for the actions, but this was not clear. There are mentions of state-action embedding which makes it seem like they are embedded to a single space?
* The results between ConDT w/o Prod and ConDT are mixed: it seems like it is not clear which method to use.

**Summary Of Recommendation:**

Decision transformers are a promising sequence modeling approach to reinforcement learning and this work suggests an improved version with a contrastive loss to separate state, actions for different returns. It would be useful for the community to reproduce this work on additional benchmarks, especially real robot setups. The contrastive loss introduced is not revolutionary and likely a large transformer has the modeling capacity to accomplish, however, in simulated benchmarks shown the results are convincing improvement over DT.

---

> ### Author Response · Authors · 2022-08-27
> **Author Responses to Reviewer JfJf**
>
> Thank you! We appreciate your time and dedication. Please find below our point-by-point responses to your comments:
>
> - **Real Robot Experiments**: Thank you! We agree that including real robot experiments is a necessity and doing so is our intention. As such, we added a new experiment on the Adroit Robotic HandGrip Environments, wherein a 24 degree-of-freedom robotic hand is trained on four different tasks, such as opening a door and hammering a nail. All tasks have a reward function proportional to the percentage of the task completed, so if the hand closes a door 50% of the way, half the maximum reward is given. Additionally, all trajectory was generated by a human operating an Adroit hand in each of the four scenarios. Therefore, success of ConDT in the simulation is in close correlation with its applicability to a real-world Adroit hand. Our new results (Quantitative Results Table 3 and Video Demonstration uploaded as a Supplementary File) demonstrate the effectiveness of our method and applicability of ConDT to real robots. We also added a new paragraph to the text describing how ConDT and similar offline RL methods are relevant methods for robot learning and how to deploy such methods on real physical robots.
>
> - **Setups where Markov-Assumption Clustering Might Hurt**: This question is related to your last comment that “the results between ConDT w/o Prod and ConDT are mixed.” We discuss this point in the results for the Atari experiments (lines 244 to 258) and notice that sometimes ConDT w/o Prod outperforms ConDT. We theorize the reason for this is because Atari is one of those environments that is not Markovian, and the return at a given step is tied to the returns at previous steps. In ConDT w/o Prod, we still apply the SimRCRL loss, but we feed the return embeddings into the input sequence of GPT, unlike ConDT where we only feed the return-encoded, state and action embeddings. Attention allows the return information from preceding timesteps to be integrated into the current step. Thus, we hypothesize ConDT w/o prod might be more appropriate for non-Markovian timesteps.
>
> - **Some Details of the Transformer Architecture**: In our experiments, DT and ConDT have the same number of parameters; the only difference is that on ConDT we have a linear projection layer that combines the state and action embeddings into a state-action embedding for the SimRCRL loss. When we did our ablations, we used the same linear projection layer between ConDT and DT to generate the visualization of the state-action embeddings. We agree that a large enough transformer might implicitly model this separation and testing different transformer sizes is definitely a facet of future work – we decided to use the same GPT configurations used in the original DT paper. However, there is no guarantee of this separation, and one way we found to consistently guarantee this separation was to use our direct distance optimization objective, SimRCRL
>
> - **Embeddings for States and Actions**: For the SimRCRL we use a linear projection layer to “compress” the individual state and action embeddings, into a state-action embedding. This compressed embedding is not used for anything except the SimRCRL loss – we still feed state and action embeddings into DT
>
> - **Results Between ConDT w/o Prod and ConDT are Mixed**: •	Please see our discussion above to your second comment
>
> ## Issues:
>
> - Thank you. This is definitely a facet of the future work, and we will include such ablation experiments in our future extended versions of the paper
>
> - Please see our discussion above to your second comment
>
> - If we assume rewards are either (0, 1), then we theorize that the performance of the DT should not be affected. The performance of DT is tied to the returns provided in the offline dataset, and the returns with binary rewards will still be informative to the DT. If these rewards are sparse, then we still theorize our methods will show promise, because in the original DT paper, they tested on a sparse-reward environment, Reacher2D. We did not benchmark in Reacher because the dataset was not publicly available.
>
> We hope that our responses and added simulated robotic handgrip experiment, discussions, and details were satisfactory. If so, we would appreciate if the respected reviewer considered increasing their score. Also, if there’s any further questions or concerns, we would be more than happy to address them.

---

### Meta-Review · Area_Chair_veKg · 2022-08-14

**Recommendation:** Accept (Poster)
**Confidence:** 3

**Metareview:**

Phase 1:

Strengths:
The submission is intuitively and clearly written. The application of contrastive losses to decision transformers is clear, well presented and provides convincing improvements on some established benchmarks. Provided ablations are helpful.

Weaknesses:
The combination of DT and SIMCLR loss can be seen as straightforward, which would be fine with strong empirical improvements. While these are convincing on Atari, improvements to the simpler DT+Prod baseline in Gym are limited. Further, common offline RL baselines are missing and the comparison focuses on variations of DT as baseline.
No real robot experiments are performed and missing clarity regarding the path to application was emphasized (maybe a discussion of computational feasibility would be helpful here). Additional details are missing to improve the presentation (e.g. embeddings, size comparisons between DT and ConDT, increased computational costs).

Phase 2:

The feedback has been a borderline case with positive and negative feedback. While it is seen as well written and intuitive, the contributions are marked as straightforward while improvements are limited. During the rebuttal, the authors have been able to convince all reviewers to change their ratings to weak accepts. Main criticisms towards comparing against offline RL state-of-the-art methods remain but the connection towards robotics has been strengthened with additional empirical results.

The paper remains a very close borderline case which should be extended with additional baselines instead of purely focusing on the decision transformer. In the end, my recommendation follows the reviewer's suggestion of acceptance because of the added simulated robot experiments. However, I strongly suggest the addition of further baselines for a camera-ready version if accepted.

**Best Paper Nomination:**

No

---

> ### Author Response · Authors · 2022-08-27
> **Author Responses to Meta Comments -- Part I**
>
> We appreciate all the reviewers and the meta-reviewer for taking the time to review our paper and for their constructive comments and feedback. We would like to address the summary points mentioned by the meta-reviewer:
>
> - **DT+Prod Baseline**: We believe that there is a misunderstanding. The baseline mentioned by the respected reviewer (i.e., DT+Prod) is still created by us, as an incremental step towards the final variant of our method (i.e., an ablation) and is not the naïve DT. ConDT variants (with the contrastive loss), which are built with and without the Prod step, substantially outperform the base DT. Moreover, ConDT outperforms DT in Gym by 10%; the reason that is not as high as the performance improvement in Atari is that DT was already performing close to human-target performance prior to any of our optimizations. Using ConDT just improved the performance of Gym experiments even more to bring return results closer to human-target returns. In terms of methodology, we agree that our methodology is an incremental improvement from DT and RCRL individually (most research is incremental); the fact that DT is return-conditioned and applying a return-based contrastive loss was the main part of the novelty which has not been addressed before. SimRCRL is also an improvement of RCRL because it directly optimizes our optimization objective of separating the distance between state-action embeddings. Despite the seemingly straightforward derivation of ConDT, our approach substantially outperforms prior work.
>
> - **Other Offline RL Baselines**: We agree that this addition could add to the value of the paper. We note that the original DT was compared to the SOTA offline RL algorithms, demonstrating better or comparable performance in all scenarios. Based upon this result, we choose the DT as the state-of-the-art. As such, we only focus on showing that our work can outperform the DT as the SOTA, which was already shown to be better than other offline RL baselines. We will include other baselines in future extensions of our work.
>
> - **Applicability to Robot Learning**: Our approach, ConDT, and other similar offline RL methods are relevant to real physical robot learning, just as LfD writ large are relevant (see [1-2] as instance). A key difference is that many LfD methods are in fact online and would thus likely need more environment rollouts. It is true that offline RL methods are restricted to the dataset and have limited exploration capability. Nevertheless, many robotics applications may not allow for exploratory actions, such as in robotic surgery. Therefore, we believe while offline RL and LfD are both highly applicable to robot learning, they can have different use-cases. We agree that including real robot experiments is a necessity and doing so is our intention. As such, we added a new experiment on the Adroit Robotic HandGrip Environments, wherein a 24 degree-of-freedom robotic hand is trained on four different tasks, such as opening a door and hammering a nail. All tasks have a reward function proportional to the percentage of the task completed, so if the hand closes a door 50% of the way, half the maximum reward is given. Additionally, all trajectory was generated by a human operating an Adroit hand in each of the four scenarios. Therefore, success of ConDT in the simulation is in close correlation with its applicability to a real-world Adroit hand. Our new results (Quantitative Results Table 3 and Video Demonstration uploaded as a Supplementary File) demonstrate the effectiveness of our method and applicability of ConDT to real robots. We also added a new paragraph to the text describing how ConDT and similar offline RL methods are relevant methods for robot learning and how to deploy such methods on real physical robots.
>
> [1] Sinha, Samarth, Ajay Mandlekar, and Animesh Garg. "S4RL: Surprisingly simple self-supervision for offline reinforcement learning in robotics." Conference on Robot Learning. PMLR, 2022.
>
> [2] Chen, Letian, Rohan Paleja, and Matthew Gombolay. "Learning from suboptimal demonstration via self-supervised reward regression." Conference on Robot Learning. PMLR, 2021.
>
> - **Missing Details** : Thank you! We revised and added all the missing details. We added new experiments focused on robot learning tasks, demonstrating applicability of ConDT for task learning in robots. We also added new analysis on execution gains, comparing the training and validation times required for ConDT and other baselines We hope the modifications to the text also improved the readability and understandability of ConDT. Additionally, we prepared a video demonstration of our experiments on the Adroit Robotic HandGrip domain as a supplementary material.

---

> > ### Author Response · Authors · 2022-08-27
> > **Author Responses to Meta Comments -- Part II**
> >
> > We hope that our responses and added simulated robotic handgrip experiment, discussions, and details were satisfactory (all revisions are marked red in the new paper and supplementary material uploads). If so, we would appreciate if the respected reviewers considered increasing their score. Also, if there’s any further questions or concerns, we would be more than happy to address them